# The Short-Term Effects and Tolerability of Low-Viscosity Soluble Fibre on Gastroparesis Patients: A Pilot Clinical Intervention Study

**DOI:** 10.3390/nu13124298

**Published:** 2021-11-28

**Authors:** Harsha Suresh, Jerry Zhou, Vincent Ho

**Affiliations:** 1School of Medicine, Western Sydney University, Campbelltown, NSW 2560, Australia; 17271790@student.westernsydney.edu.au (H.S.); v.ho@westernsydney.edu.au (V.H.); 2Gastrointestinal Motility Disorders Unit, Western Sydney University, Campbelltown, NSW 2560, Australia; 3University Medical Clinic of Camden & Campbelltown (UMCCC), Campbelltown, NSW 2560, Australia

**Keywords:** soluble fibre, gastroparesis, motility disorder, ANMS GCSI-DD, glucose regulation

## Abstract

Gastroparesis is a motility disorder that causes severe gastric symptoms and delayed gastric emptying, where the majority of sufferers are females (80%), with 29% of sufferers also diagnosed with Type-1 or Type-2 diabetes. Current clinical recommendations involve stringent dietary restriction and includes the avoidance and minimization of dietary fibre. Dietary fibre lowers the glycaemic index of food, reduces inflammation and provides laxation. Lack of dietary fibre in the diet can affect long-term gastrointestinal health. Our previously published rheological study demonstrated that “low-viscosity” soluble fibres could be a potentially tolerable source of fibre for the gastroparetic population. A randomised controlled crossover pilot clinical study was designed to compare Partially-hydrolysed guar gum or PHGG (test fibre 1), gum Arabic (test fibre 2), psyllium husk (positive control) and water (negative control) in mild-to-moderate symptomatic gastroparesis patients (requiring no enteral tube feeding). The principal aim of the study was to determine the short-term physiological effects and tolerability of the test fibres. In *n* = 10 female participants, post-prandial blood glucose, gastroparesis symptoms, and breath test measurements were recorded. Normalized clinical data revealed that test fibres PHGG and gum Arabic were able to regulate blood glucose comparable to psyllium husk, while causing far fewer symptoms, equivalent to negative control. The test fibres did not greatly delay mouth-to-caecum transit, though more data is needed. The study data looks promising, and a longer-term study investigating these test fibres is being planned.

## 1. Introduction

Gastroparesis is a chronic gastrointestinal disorder where sufferers experience reduced stomach functionality with no mechanical obstructions, delayed gastric emptying and debilitating symptoms [1]. Symptoms associated with gastroparesis include nausea, retching, post-prandial fullness, early satiety, bloating and distention [2]. In Australia, there are estimated 120,000 gastroparesis sufferers, an estimated 0.5% of the total population, with the majority of sufferers (80%) being female [3].

The onset of gastroparesis typically begins in adolescence, with a substantial number of patients developing gastroparesis after major gastrointestinal trauma such as infection, surgery, or physical injury [4,5]. Idiopathic gastroparesis has significant reported co-morbidity with Type-1/Type-2 diabetes, with 29% patients suffering from a type of gastroparesis known as “diabetic gastroparesis” [6]. A smaller subset of gastroparesis is associated with neurological and vascular disorders such as Parkinson’s disease, Ehlers-Danlos syndrome and multiple sclerosis [7,8,9]. Since gastroparesis currently has no known cure, gastroparesis symptoms are managed through a combination of dietary modification and medication [10]. 

Dietary fibre is a critical component of a balanced and healthy diet [11]. The two primary types of dietary fibre are soluble and insoluble, with both these types having an effect on gastric emptying [12]. The major benefits of soluble dietary fibre include blood glucose regulation, reduced risk of post-prandial hyperglycaemia & hyperosmolar hyperglycaemic nonketotic syndrome (HNNS), and a prominent role as a glycaemic index (GI) lowering macronutrient [13,14,15,16]. The laxative effects of soluble fibres are more variable with respect to constipation, diarrhoea, and IBS [17,18], and these effects depend largely on chemical structure and rheological behaviour [19,20]. Soluble dietary fibres are also known to release beneficial short-chain fatty acids (SCFA) such as acetate, propionate, and butyrate into the colon through gut microbiome fermentation and this has been proven to lead to better colonic health, especially in patients with Type-1/Type-2 diabetes [21,22]. Supplementation of the soluble dietary fibre partially-hydrolysed guar gum (PHGG) alongside routine medication has been reported to reduce small intestinal bacterial overgrowth (SIBO) [23]. Dietary recommendations for healthy adults encourage the consumption of 25–30 g of dietary fibre daily, as a key component of a healthy diet [11,24,25]. Despite the known benefits, the current dietary recommendations for gastroparesis patients suggests the avoidance or minimisation of all forms dietary fibre [26]. The restrictions increase as gastroparesis patients begin to experience chronic delayed gastric emptying, worsening symptoms and disease progression [27]. Although the dietary restriction of fibre can temporarily relief to gastroparetic symptoms, it can have detrimental effects on a patient’s long-term gastrointestinal and metabolic health. 

There is a pressing need for dietary fibre alternatives in mild-to-moderate symptom gastroparesis patients, whose disease condition has not progressed to the stage where they require enteral tube feeding for nourishment [28]. Not all soluble dietary fibres are the same, and properties such as chemical structure and viscosity affect the physiological effects they may produce in the stomach. Our previous study investigated the chemical and rheological properties of ten soluble fibres in vitro in simulated gastric fluid [29]. We determined that the linear, low molecular weight partially-hydrolysed guar gum (PHGG) and the branched, medium molecular weight gum Arabic soluble fibres retained their “low-viscosity” characteristics during simulated digestion. The next logical step was to evaluate their performance in individuals with gastroparesis in vivo in a clinical setting.

The principal aim of this pilot clinical study was to determine the short-term blood-glucose regulation and tolerability of PHGG and gum Arabic in mild-to-moderate symptom gastroparesis patients. The primary endpoint of this study was to establish the blood glucose profile following a glucose challenge and tracking changes in patient-rated gastroparesis symptoms. The secondary endpoint was the measurement of the “starting” or “reaching” of mouth-to-caecum transit. 

The test fibres were compared against a negative control (water) and positive control (psyllium husk) in their abilities to modulate blood glucose concentration following a glucose challenge. “High-viscosity”, high molecular weight psyllium husk is a commonly prescribed fibre supplement, and its physiological effects have been investigated extensively in clinical studies [30,31]. Similarly, there have been clinical investigations of the physiological effects of low-viscosity PHGG [32,33] and gum Arabic [34,35], respectively. Pin-prick blood glucose concentrations were recorded at 30-min intervals during the study. Gastroparesis symptoms were monitored using a patient-rated validated questionnaire, The American Neurogastroenterology and Motility Society Daily Diary (ANMS GCSI-DD) for gastroparesis patients [36,37,38,39]. Breath testing was used as an estimator of the “reaching” or “starting” of oral-caecal transit [40], in order to identify alterations following dietary fibre consumption. 

## 2. Materials and Methods

### 2.1. Study Design & Premises

The pilot clinical study was designed as a randomised controlled crossover study with four arms that studied the short-term effects of low-viscosity dietary fibre meals during a 3-h period in patients with mild-to-moderate symptom gastroparesis. The randomised controlled crossover study design in a small number of participants was ideally suited as a baseline for future investigations. By studying three different fibres and the negative control in the same participant (crossover), normalizing the collected data (thereby reducing variability), the blood glucose changes, and symptom effects caused by the dietary fibre interventions become comparable both within the same participant (internal control) and among other participants (mild-to-moderate symptom gastroparesis patients).

The study lasted four sessions for each participant with water (negative control) as the first meal. Participants waited for a minimum 1-week in-between each session, assigned as the washout period between test meals [41]. In weeks 2, 3 & 4 the intervention order of the dietary fibre test meals of psyllium husk (positive control), PHGG (test fibre 1) and gum Arabic (test fibre 2) were randomised. Participants were given 10 g of a dietary fibre test meal and were allowed to end the meal when their tolerable meal amount was reached. The consumed amount of each meal varied from participant to participant and was measured. The PHGG (Sunfiber™) supplement was purchased from Healthy Origins, Australia. Gum Arabic was purchased from New Directions, Australia. Psyllium husk was purchased from SF Health Foods, Australia. The clinical ethics for this study was approved by the Western Sydney University Human Ethics Resources Committee (HREC) with the ethics reference number H12254. The workflow of the clinical study progression is shown in Figure 1. 

The participants’ weight and height were measured using a Marsden M-100 clinical scale from Marsden, UK. The glucose measurements were performed using a FreeStyle Optium Neo™ blood glucose and ketone probe meter from Abbott Laboratories, USA. Test strips, glucose and ketone calibration solutions for the probe meter were also purchased from Abbott Laboratories, USA. Participants were given a glucose challenge (unflavoured 50 g in 300 mL) Glucoscan™ glucose drinks from Cleanway Daniels, Australia. In addition, Actilax™ (lactulose, 3.3 g/5 mL) (Chemist’s Warehouse, NSW, Australia) was used to determine the “reaching“ or “starting” of mouth-to-caecum gastric transit through its initial breakdown and gas production in the caecum. The symptom severity was measured using the ANMS GCSI-DD validated survey [42], which is recognised by the Food & Drug Administration (FDA) as one of the most rigorous measures of patient-rated gastroparesis symptoms [39]. 

The breath hydrogen (H_2_) and methane (CH_4_) values monitored for indications of the start of mouth-to-caecum transit, were measured using a Quintron Breath Tracker™ instrument purchased from Quintron, USA. The 750 mL breath test bags, the injection tube Drierite™ (98% CuSO_4_, 2% CoCl_2_) desiccant, the instrument drying desiccant SivRite-4™, and the QuinGas-3™ (150 ppm H_2_, 75 ppm CH_4_, 6.2 ppm CO_2_) calibration gas were also purchased from Quintron, USA. The instrument was calibrated and cycled at full and half volume before use. Breath-blows with CO_2_ concentration higher than 6.0 ppm were discarded and the breath-blows were repeated to ensure a valid result. Pin-prick blood glucose, patient-rated ANMS GCSI-DD symptoms and breath test measurements were taken at baseline (0 min, pre-glucose challenge), and at 30 min intervals after test meal consumption and post-glucose challenge. The study was conducted at the designated clinical rooms of the MacArthur Clinical School, Western Sydney University (WSU), in Campbelltown, NSW 2560, Australia. 

### 2.2. Participant Diagnosis & Eligibility 

All potentially eligible participants (both female and male) in this study were diagnosed within the preceding five years. Gastric scintigraphy or a gastric emptying study is currently considered the “gold standard” for gastroparesis diagnosis by physicians [43]. Eligible participants in this study met the scintigraphic criteria for gastroparesis diagnosis, which is defined as the minimum of 10% retention of solid food contents at 4 h after the consumption of a Tc-99m radio-labelled meal [43]. It should be noted that all participants in this clinical study had completed a scintigraphy study no more than 12 months prior to their participation, in order to either confirm or re-confirm the severity of their delayed gastric emptying and gastroparesis symptoms. The eligible participants’ gastroparesis-related symptoms during the period of the clinical study were being managed with routine medications prescribed for gastroparesis patients [44]. 

### 2.3. Inclusion & Exclusion Criteria and Risks

The inclusion criteria permitted participants aged 18 years or older (adults), who were able to provide written consent. Participants diagnosed with idiopathic gastroparesis or diabetic gastroparesis (Type-1/Type-2) were eligible to participate in the study. Participants excluded from the study included pregnant women, people who could not provide self-consent, and those diagnosed with co-morbidities related to gastroparesis excepting Type-1/Type-2 diabetes mellitus. Participants diagnosed with celiac disease were also excluded, due to the use of wheat-based products in this clinical study [45]. Individuals with severe gastroparesis requiring enteral feeding tubes were excluded from this study due to the risk of tube blockage. 

### 2.4. Participant Recruitment 

Participants were recruited through a recruitment poster on the Gastroparesis Australia support group website. Potential participants were also recruited though word-of-mouth information provided by their physician. Interested participants were screened through an online questionnaire on the Gastrointestinal Motility Disorders Unit website. After screening, investigators made contact to explain the study and provided them with the Participant Information Sheet. The tests were conducted at MacArthur Clinical School, Campbelltown, NSW, Australia, where the participant was provided with the consent form before commencement. Overall, *n* = 10 participants were recruited into this clinical study, all of them female. General information about study participants is provided in Table 1. 

### 2.5. Data Curation & Analysis 

Microsoft Excel (Office 2016) was used to analyse and generate usable data from collected pin-prick blood glucose measurements, patient-rated ANMS GCSI-DD symptom scores and breath test measurements. Dataset normalization of the 4-week study of each participant enabled direct comparisons of changes in blood glucose and symptoms among all participants in the dataset. The statistical power of normalization enhanced the data analysis, since the effect of major variables such as co-morbidities, medication and gender were carefully controlled in this study. It should also be noted that participants with co-morbidities typically associated with gastroparesis other than Type-1/Type-2 diabetes were excluded, and routine medications for each participant was maintained across all four test weeks during the fasting period (12 h prior to the start of each test session), further minimizing dataset variability.

Normalization of the collected glucose data was performed by subtracting the measured, pre-meal blood glucose (at 0 min) from the blood glucose measurements of subsequent data points (post-meal, 30 min–180 min). Similarly, the ANMS GCSI-DD scores were also normalized using “Mean of Mean” ANMS GCSI-DD scores obtained by subtracting the baseline symptom scores (0 min) from summed mean scores (30 min–180 min) for each symptom post-meal. Standard deviation (SD) values were calculated for each mean normalized data point. For clarity and brevity, the 9-symtpom GCSI scorecard was truncated into three representative symptom composites, which conforms with published reports using the validated ANMS GCSI daily-diary [39]. The three composites used in this analysis are Composite (1) for Nausea/Retching related symptoms, Composite (2) for post-prandial fullness (PPF)/Early Satiety related symptoms and Composite (3) for Bloating/Distension related symptoms. 

The Food & Drug Administration (FDA) has stated in a recent guidance report (2018) that “Bloating” as a symptom can be deemed too similar to “post-prandial fullness” and can be removed the GCSI daily-diary [46]. For the purposes of this study, the patient-rated “Bloating” related Composite (3) was included as an exploratory item. All stated two-tailed, homoscedastic *p*-values in this text were generated using the Student’s t-test in R version 4.0.3 and R Studio (Graphic User Interface). The baseline “stats”, “utils”, “methods”, “graphics”, “datasets” and “grDevices” R packages were used. 

## 3. Results 

### 3.1. Blood Glucose Monitoring 

The mean normalized blood glucose peak values and the interval area under the curve (iAUC) for the ten participants in the study are summarised in Table 2. It is very important to note that participants on average, were only able to tolerate 4.13 g of a psyllium husk meal as opposed to 7.99 g of a PHGG meal and 7.57 g of a gum Arabic meal. The participants in general, were not able complete their assigned dietary fibre meals (10 g) and this played an important role in determining the usefulness and relevance of a soluble dietary fibre to a gastroparesis patient.

It can be seen in Figure 2 that the mean normalized blood glucose peaked at 60 min for water (5.9 mMol/L). This can be contrasted to the lower mean normalized blood glucose peaks at 30 min for the test fibres PHGG (3.9 mMol/L), gum Arabic (4.1 mMol/L) and test fibre psyllium husk (3.9 mMol/L). When PHGG and gum Arabic are compared to the positive control meal psyllium husk (3.9 mMol/L) at 30 min, no significant increase or decrease in blood glucose is observed (*p* > 0.05). Following blood glucose peaks at the 30–60 min time-points, mean blood glucose begins to gradually reduce during the 90–120 min interval. The normalized blood glucose values at 90–120 min time-points range from 0.7–2.6 mMol/L, with minor differences observed between the test fibres and the negative control. At the 150–180 min time-points, the test fibres (PHGG & gum Arabic) return to baseline faster (at 150 min) as opposed to the slower return to baseline observed for the negative control (water) and positive control (psyllium husk) at 180 min. The test fibres (PHGG & gum Arabic) “dip” slightly below baseline blood glucose at 180 min (−0.7 mMol/L). While severe hypoglycaemia (blood glucose < 2.0 mMol/L) can be a significant risk in diabetic gastroparesis patients [47], the lowest measured blood glucose value among the (*n* = 10) participants was 2.9 mMol/L, which occurred during a participant’s water meal (negative control) session. None of the participants reported any symptoms typically related to hypoglycaemia [48].

The low-GI capabilities of all three test fibres are summarised by the 1-h iAUC values of PHGG (169.50 mMol·min/L), gum Arabic (184.93 mMol·min/L) and psyllium husk (169.13 mMol·min/L) when compared to water (241.83 mMol·min/L), as shown in Table 2. It should also be noted that total 2-h iAUC values of the dietary fibres PHGG (309.00 units), gum Arabic (322.29 units) and psyllium husk (325.50 units) are still very similar to each other and are not significantly different (*p* > 0.05). This 1–2-h post-prandial period is less important since blood glucose typically returns to fasting (overnight, 12 h) level post-peak, in the 2–3-h monitoring period [49]. 

At the 3-h time point, as expected, all test meals returned to their pre-meal baseline with a negligible change in iAUC. Interestingly, both the test fibres PHGG and gum Arabic return to baseline levels at 150 min in contrast to the 180 min mark observed for psyllium husk and water. This observed discrepancy was not significant when compared statistically (*p* > 0.05), possibly due to the small sample size (*n* = 10). In *n* = 5 participants the “monophasic” single glucose peak pattern of a water meal (negative control) was converted into “biphasic” double glucose peaks when they were given the dietary fibre test meals of PHGG, gum Arabic and psyllium husk (positive control). 

### 3.2. ANMS GCSI-DD Monitoring 

The summarised, mean normalized ANMS GCSI-DD scores across ten participants are shown in Table 3. Eight significant correlations, in total, are observed when comparing the composite symptom scores among ten participants. There are significant comparisons when comparing individual symptoms across the four test meals, but these are truncated into the composite scores due to the limited sample size of participants (*n* = 10). Significant correlations among individual symptoms are also shown in Table 3 but are not discussed in any great detail. 

When Nausea-related (1) composite scores are compared, it is clear that the negative control water (−0.22) slightly reduced nausea when compared to the baseline measurement. The positive control psyllium husk (0.51) displays significantly increased Nausea-related symptoms compared to water (−0.22), with *p* = 0.05. This result is entirely expected since psyllium husk is a “high-viscosity” dietary fibre. What was unexpected was the noticeable increase in Nausea-related symptoms from the PHGG meal (0.85), although this increase is not significant when compared to water (−0.22), with *p* > 0.05. It is important to note that there is a large standard deviation (SD) of 1.30 for the Nausea-related composite of PHGG. This informs us that some participants experienced increased Nausea-related symptoms after a PHGG meal, while others did not, resulting in the noticeably large standard deviation.

When the mean of the “highest” or “worst” patient-rated symptom scores (“Mean of Max” scores, not shown in Table 3) of PHGG (1.88) and gum Arabic (0.71) for the “Nausea” symptom are compared, a statistically significant difference is observed (*p* < 0.05). This indicates that for PHGG, the “Nausea” symptom is disproportionately causing an increase in the Nausea-related composite with “Vomiting” and “Retching” contributing minimally. Gum Arabic (0.07) caused little to no increase in Nausea-related symptoms when compared to the negative control water (−0.22), with *p* > 0.05.

Four significant differences are observed when comparing the PPF-related (2) scores among the four test meals. In comparing PPF-related symptoms, the negative control water meal (−0.31) was significantly lower than positive control psyllium husk (0.91) with *p* = 0.01. The water meal (−0.31) also had lower PPF-related symptoms when compared to gum Arabic (0.57), with *p* = 0.01. PHGG (0.15) shows little significant change when compared to the negative control water (−0.31), with *p* > 0.05. Both gum Arabic (0.57) and PHGG (0.15) demonstrated lower PPF-related symptoms when compared to positive control psyllium husk (0.91), with *p* ≤ 0.05. This data reveals that when PPF-related symptoms like “Stomach Fullness”, “Early Satiety”, “Post-prandial Fullness” and “Loss of Appetite” are considered, the effects of a PHGG meal was comparable to drinking a “glass of water”.

When Bloating-related composite scores are compared, three significant changes are observed. A highly significant difference is observed when negative control water (−0.17) is compared to positive control psyllium husk (1.40), with *p* < 0.01. This result was entirely expected, since the higher-viscosity positive control meal was expected to increase Bloating-related symptoms, especially compared to the negative control. PHGG (0.40) demonstrated a slight increase in Bloating-related symptoms, although no significant correlations are observed when compared to other meals, with *p* > 0.05. 

Gum Arabic (0.31) displays significantly reduced Bloating-related symptoms when compared to the positive control psyllium husk (1.40), with *p* < 0.05. Interestingly, when gum Arabic (0.31) is compared to water (−0.17), a correlated symptom increase can be observed, with *p* = 0.05. This correlation might have occurred due to the consistently higher “Bloating” scores of gum Arabic (0.50) compared to PHGG (0.44) and water (−0.17) across all *n* = 10 participants. Even still, the observed Bloating-related symptom increases are very small in comparison to the symptom increase observed in the positive control. 

### 3.3. Breath Hydrogen/Methane Monitoring

Among ten participants, only five showed any signs of “reaching” or “starting” mouth-to-caecum transit within the 180 min time frame, which can be identified as a greater than 6 ppm but lesser than 12 ppm increase in breath hydrogen and methane (H_2_ + CH_4_) levels from the baseline measurement taken at 0 min. This is the standard ppm guideline range for the start of mouth-to-caecum transit of the indigestible lactulose sugar. It should be noted that the breath tests measured the “reaching” or “starting” time-point of gastric transit not its completion or “ending” (≥90% gastric emptying) time-point. This result is not entirely unexpected, since most gastroparesis patients show signs of delayed mouth-to-caecum transit in the longer, 4-h gastric scintigraphy studies [50]. 

One participant was able to show signs of mouth-to-caecum transit commencement of all four meals, water (at 150 min), PHGG (at 180 min), gum Arabic (at 180 min) and psyllium husk (at 180 min). Two other participants showed signs of mouth-to-caecum transit of the water meal at 150/180 min with transit for PHGG and gum Arabic starting at 180 min. Two participants showed signs of mouth-to-caecum transit of PHGG at 180 min but did not show signs of transit for water, gum Arabic or psyllium husk within 180 min. The participant who was able to show signs of mouth-to-caecum transit for all four test meals was the only one that showed any signs of psyllium husk transit (at 180 min). 

Three participants co-produced H_2_ and CH_4_, while seven participants exclusively produced H_2_ with < 1 ppm CH_4_ produced across all measurements. Two participants produced greater than 12 ppm H_2_ + CH_4_, a positive indicator for SIBO, but these specific participants were already diagnosed with SIBO (Small Intestinal Bacterial Overgrowth) [51] by their physician. It must be noted that the breath test was not used to diagnose SIBO among any of the participants. The breath testing information is summarised in Table 4. 

## 4. Discussion

In summary, the short-term physiological effects of soluble dietary fibres were comprehensively studied in ten mild-to-moderate symptom gastroparesis patients. This pilot clinical study demonstrates that low-viscosity soluble dietary fibres PHGG, and gum Arabic have blood-glucose regulation properties and are tolerable for patients with mild-to-moderate symptom gastroparesis.

The blood glucose data suggests that all three dietary fibres, in general, are able to “hold” more glucose from the glucose challenge during the 1st post-prandial hour compared to the no-fibre water meal. The iAUC results suggest that PHGG and gum Arabic are comparable to psyllium husk at blood glucose regulation. Lower iAUC values were observed for all three dietary fibres compared to the negative control water, due to their ability to “trap” a lot of glucose until the further release of glucose for absorption in the small intestine, beyond the 3-h monitoring period of this clinical study [52]. This is an important development, since it has been shown in previous randomised clinical studies that consumption of “high-viscosity” dietary fibres such as guar gum, β-Glucan and psyllium husk have been shown to result in improved blood glucose regulation and reduced insulin resistance in Type-2 diabetic patients [53]. The follow-on benefit of this regulation is improved long-term insulin control, reduced risk of post-prandial glycemia, significantly reduced inflammation, and better metabolic health [54,55,56,57].

A “monophasic to biphasic” peak transition was observed in five participants after the consumption of dietary fibres used in this study, irrespective of type. This phenomenon might have occurred due to the “two-phase” controlled release of insulin, which is closely regulated by single β-cells, pancreatic islets, and the whole pancreas [58]. The first release phase of insulin occurs within the 10-min post-prandial time-period followed by a long second phase of 2–3-h [59]. Even though the “monophasic to biphasic” trend is not significant enough to be apparent in Figure 2, its observance in half the participants of this study is a positive development, since biphasic peaks are associated with higher insulin sensitivity, and the lowered incidence of impaired glucose tolerance and metabolic syndrome [60].

While the blood glucose effects of all three dietary fibres are very similar, it is important to note that participants were able to tolerate much less psyllium husk in comparison to PHGG and gum Arabic. A smaller meal of psyllium husk was able to have an effect size similar to PHGG and gum Arabic, despite its lower tolerability. All three soluble dietary fibres (unlike insoluble dietary fibres) are able to form gels in the pH 2.0–4.0, HCl rich environment of the adult human stomach [12] and are successfully able to “trap” and gradually release glucose through entanglement and molecular interactions [61]. High-viscosity psyllium husk’s outsized effect was likely due to its better gelation ability in vivo compared to low-viscosity PHGG and gum Arabic [62]. Still, larger meals of PHGG and gum Arabic fibre will be able to release more beneficial SCFAs downstream in the small intestine where glucose and lipid metabolism are affected by gut microbiota digestion and composition [63]. 

The ANMS GCSI-DD symptom monitoring scorecard provides a great deal of insight into the symptom effects of the test fibres PHGG and gum Arabic when compared to psyllium husk (positive control) and water (negative control). Nausea-related symptoms are intimately connected to gastric motility, with symptom relief achieved through the prokinetic medication [64], and the drinking of water, especially cold water, which is known to greatly reduce nausea [65]. While it can tentatively be stated that gum Arabic is better than PHGG when Nausea-related symptoms are compared, literature-reported side effects of PHGG such as nausea, have always been incidental and not concrete [17,66]. The Nausea-related increase observed in PHGG may be related to hyperosmia (smell sensitivity) and can be overcome by selective flavour modification [67]. 

The superior performance of PHGG in the PPF-related metric compared to gum Arabic may be due to the structural chemical differences of PHGG (linear, short-chain) and gum Arabic (branched, long-chain), both of which as previously mentioned, are “low-viscosity” soluble dietary fibres [68,69]. The mechanistic causes of PPF-related symptoms in gastroparesis patients remains elusive, and speculated causes include damage to the vagus nerve, autoimmune causes or viral vectors [70]. The data presented in this study reveals that when considering PPF-related symptoms like “Stomach Fullness”, “Early Satiety”, “Post-prandial Fullness” and “Loss of Appetite”, the symptom effects of PHGG are comparable to a “glass of water” in mild-to-moderate symptom gastroparesis patients. This clinical study also demonstrates that the low-viscosity soluble dietary fibres PHGG and gum Arabic cause very little Bloating-related symptom increases, especially compared to high-viscosity psyllium husk [71]. 

When assessing the breath test results, it was apparent that some participants were able to show signs of “starting” mouth-to-caecum transit in either or both of PHGG and gum Arabic within 180 min, but delayed transit was universally observed in participants consuming the psyllium husk meal. It is well known in the literature that “high-viscosity” gel forming dietary fibres tend to increase the viscosity of stomach fluid during digestion, thereby delaying gastric emptying [72]. Based on this data, it can be tentatively stated that the low-viscosity soluble test fibres PHGG and gum Arabic do not seem to impede mouth-to-caecum transit as much as high-viscosity psyllium husk, though no definitive conclusions can be made. 

There were prominent limiting factors that affected this study. The primary limiting factor was the relatively small sample size of *n* = 10. The crossover design and the use of normalization, as previously stated, largely mitigates this limitation and provides significant statistical power in the generated dataset, despite the small sample size. While there have been previous dietary fibre clinical intervention studies with larger sample sizes (*n* ≥ 30), they involved the comparison of one or two dietary fibres, unlike the three dietary fibres simultaneously investigated in this study [30,31,32,33,34,35]. Those studies primarily dealt with gastric conditions where the physiological effects of dietary fibres have been investigated more extensively, such as constipation, diarrhoea, IBS and diabetes. Another limiting factor was the use of a glucose probe monitor instead of a continuous glucose monitor, which could have produced more discrete information. A glucose probe meter was selected due to ease of use, patient comfort and convenience in comparison to a continuous blood glucose monitor. In a potential longer-term study, continuous blood glucose measurement could be employed at the beginning and end of a monitoring period, where the observation duration might be a few months [73]. The longer-term effects on gastric emptying and digestive behaviour in vivo could also be studied using highly sensitive gastric emptying scintigraphy, post-prandial plasma paracetamol monitoring and ^13^C breath tests at the beginning and end of a future study, rather than employing the less sensitive hydrogen and methane breath test [50,74,75].

The lack of clinical studies besides a recent case study investigating dietary fibre in relation to diabetic gastroparesis, necessitated the need for rigorous and cautious data collection [76]. The encouraging performance of the low-viscosity soluble fibres PHGG and gum Arabic in this pilot clinical study can be used as the basis for a future study investigating their longer-term effects in a larger cohort (*n* ≥ 30) of gastroparesis patients. The long-term tolerability, glucose regulation benefits, metabolic effects, and the soluble fibre-mediated changes in gut microbiota composition and its associated release of beneficial SCFAs would be of significant interest for individuals with gastroparesis.

## 5. Conclusions

In summary, the data presented in this clinical study demonstrates that the “low-viscosity” soluble fibres PHGG and gum Arabic may be viable dietary supplements for mild-to-moderate symptom gastroparesis patients. Glucose monitoring indicates that PHGG and gum Arabic have GI lowering properties comparable to “high-viscosity” psyllium husk. The ANMS GCSI-DD data indicates both test fibres are more tolerable and cause far fewer symptoms compared to psyllium husk. In some participants, PHGG and gum Arabic did not impede mouth-to-caecum transit as much as psyllium husk, if the start of mouth-to-caecum transit was observed during the monitoring period. Future studies will involve the investigation of longer-term physiological and metabolic effects of both low-viscosity soluble test fibres in the diet of gastroparesis patients.

## Figures and Tables

**Figure 1 nutrients-13-04298-f001:**
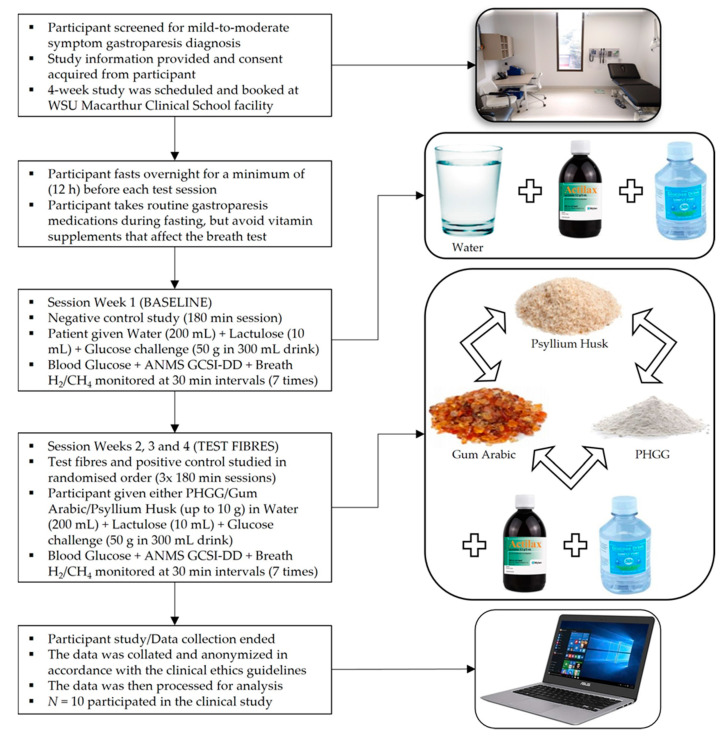
Procedural flowchart for the clinical study.

**Figure 2 nutrients-13-04298-f002:**
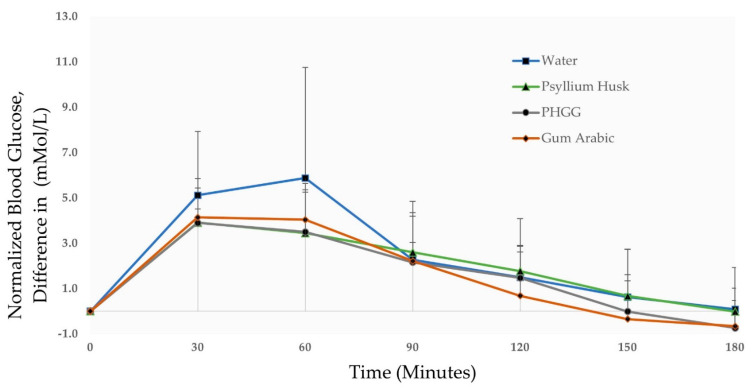
Changes in blood glucose concentration of participants (*n* = 10) given test meals of water (negative control), psyllium husk (positive control), PHGG (test fibre 1), and gum Arabic (test fibre 2).

**Table 1 nutrients-13-04298-t001:** General summary of study participants.

Demographic Information (Units)	Value
Number of Participants	10
Gender (Female/Male)	10/0
Age (Years, Mean ± SD)	38.5 ± 17.8
BMI (Index, Mean ± SD)	29.3 ± 11.7
Etiology of Gastroparesis (Idiopathic/Diabetic)	7/3

**Table 2 nutrients-13-04298-t002:** Blood glucose parameters including mean normalized blood glucose values at 30 min intervals, interval area under the curve (iAUC), and time to baseline (TTB) shown for *n* = 10 participants in the study.

Test Meal (In 200 mL Water)	Average Meal Consumed (g) ± (SD)	Mean Normalized Blood Glucose Differences (mMol/L) ± (SD), Interval Area under the Curve or iAUC (mMol·min/L), Time to Baseline (min)								
		At 30 min	At 60 min	iAUC (1 h)	At 90 min	At 120 min	iAUC (2 h)	At 150 min	At 180 min	TTB
**Water** (Negative control)	0 (0) *	5.1 (2.8)	5.9 (4.9) **	241.83	2.3 (0.8)	1.5 (1.4)	420.83	0.6 (2.1)	0.1 (1.8)	180 min
**Psyllium Husk** (Positive control)	4.13 (1.18)	3.9 (0.6) **	3.5 (1.9)	169.13	2.6 (2.2)	1.8 (2.3)	325.50	0.7 (2.1)	0.0 (1.0)	180 min
**PHGG** (Test fibre 1)	7.99 (1.92)	3.9 (1.5) **	3.5 (1.8)	169.50	2.2 (2.0)	1.5 (1.4)	309.00	0.0 (1.6)	−0.7 (1.2)	150 min
**Gum Arabic** (Test fibre 2)	7.57 (1.54)	4.1 (1.7) **	4.0 (1.6)	184.93	2.2 (2.1)	0.7 (1.9)	322.29	−0.3 (1.7)	−0.7 (0.6)	150 min

(*) All participants completed the 200 mL water meal (i.e., no test fibre). (**) Indicates the time and mean normalized concentration for the glucose peak in each test meal.

**Table 3 nutrients-13-04298-t003:** Mean normalized ANMS GCSI-DD scores across all time-points for *n* = 10 participants in the study, with Composite (1) score for (Nausea/Vomiting); Composite (2) score for (Post-prandial Fullness/Early Satiety) and Composite (3) score for Bloating/Distension.

	Mean Normalized ANMS GCSI-DD Scores ± (SD) (*n* = 10)					
Symptom Subscale (Composite No.)	GCSI-DD (Symptom)	Baseline Mean (Pre-Meal Scores) *	Water (Negative Control)	Psyllium Husk (Positive Control)	PHGG (Test Fibre 1)	Gum Arabic (Test Fibre 2)
**Nausea/Vomiting** (1)	Nausea	1.55 (1.15)	0.04 (1.22)	0.90 (0.73)	1.00 (1.41)	0.12 (0.72)
	Retching	0.45 (0.83)	−0.31 (1.03)	0.04 (0.63)	0.71 (1.47)	0.05 (0.13)
	Vomiting	0.40 (0.81)	−0.37 (1.11)	0.60 (1.16)	0.85 (1.33)	0.05 (0.13)
**Post-prandial Fullness/Early Satiety** (2)	Stomach Fullness	2.43 (1.91)	0.17 (1.22)	1.40 (1.22)	0.35 (1.04)	0.40 (0.95)
	Early Satiety *	3.35 (0.91)	−0.78 (0.97)	0.88 (0.64) **	−0.38 (1.69)	0.43 (0.98)
	Post-prandial Fullness *	3.83 (0.73)	−0.22 (0.83)	0.25 (0.46)	0.00 (0.53)	0.29 (0.76)
	Loss of Appetite	2.58 (1.82)	−0.41 (0.92)	1.10 (1.51) **	0.63 (1.02) **	1.17 (1.52) **
**Bloating/Distension** (3)	Bloating	2.10 (1.68)	−0.17 (0.70)	1.52 (1.14) **	0.44 (0.94)	0.50 (0.64) **^,^ ***
	Belly Visibly Larger	1.88 (1.77)	−0.17 (0.60)	1.27 (1.09) **	0.35 (0.71)	0.12 (0.44) ***
**Composite Scores**	Composite (1)	0.80 (1.06)	−0.22 (1.10)	0.51 (0.48) **	0.85 (1.30)	0.07 (0.29)
	Composite (2)	3.04 (1.50)	−0.31 (0.65)	0.91 (0.70) **	0.15 (0.75) ***	0.57 (0.64) **^,^ ***
	Composite (3)	1.99 (1.68)	−0.17 (0.56)	1.40 (1.09) **	0.40 (0.82)	0.31 (0.48) **^,^ ***

(*) Reported mean baseline scores (0 min) are patient-rated (*n* = 10), before each study session and not normalized, with early satiety and post-prandial fullness measured once immediately after the meal. (**) Indicates significant difference (*p* ≤ 0.05) vs water (negative control). (***) Indicates significant difference (*p* ≤ 0.05) vs psyllium husk (positive control).

**Table 4 nutrients-13-04298-t004:** General data produced by the breath test measurements.

Demographic Information	Participants (out of 10)
Number of Participants	10
H_2_ producer	7/10
H_2_ + CH_4_ producer	3/10
SIBO (Already diagnosed)	2/10
**Mouth-to-Caecum** **Reaching/Starting Transit** **(6 < ppm < 12, within 180 min)**	**Participants (out of 10)**
Any sign of transit	5/10
Water meal	3/10
Psyllium Husk meal	1/10
PHGG meal	4/10
Gum Arabic meal	2/10

## Data Availability

The anonymised clinical data files underpinning the research presented in this article are available from the corresponding author upon request.

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
