# Peer review of "The Short-Term Effects and Tolerability of Low-Viscosity Soluble Fibre on Gastroparesis Patients: A Pilot Clinical Intervention Study"

_nutrients, 2021, doi:10.3390/nu13124298_

Round 1

Reviewer 1 Report

A welcome and well written paper on an important topic with a big need for future research.

Minor points only

I don't think this recommendation in the abstract is true: 'the avoidance of all dietary fibres'. Not possible!

Abstract (and a little elsewhere), needs some revision over use of upper and lower case first letters (eg 'Negative control')

Introduction, suggest add pain as a symptom. Appreciate it's not in the scoring tool used, but that is a flaw in the tool as it is such a common symptom.

Line 43, reword 'with soluble fibres being easier to gastrically empty'

Aims clear. Methods appropriate and very well described. Not sure the H2/CH4 BTs add much really though, and the inclusion of SIBO patients was curious, was this discovered post hoc? 

Additional tests to measure gastric emptying (13C  breath tests, plasma paracetamol) would have strengthened the study more than the OCTT part.

What was the method of randomisation?

Table 1, suggest report age and BMI to only 1 d.p. 

Results are well described and interesting, though please remove word 'neatly' on line 213 though. Also I don't think it's a paradox that the BG was lowest after water.

Discussion is balanced and fair. Not sure should yet be suggesting use of gastric alimetry as it is a new commercial device.

Author Response

Q: I don't think this recommendation in the abstract is true: 'the avoidance of all dietary fibres'. Not possible!

R: The text in the abstract has been corrected to “minimization and avoidance of dietary fibre” (Line 13-14). This change now conforms with the explanation in the text (Line 56-58).

Q: Abstract (and a little elsewhere), needs some revision over use of upper and lower case first letters (e.g., 'Negative control')

R: Lower case bracket letters are now used for ‘negative control’; ‘positive control’; ‘water’; ‘psyllium husk’; ‘gum Arabic’. Upper case used for listed terms in Tables / Figures.

Q: Introduction, suggest add pain as a symptom. Appreciate it's not in the scoring tool used, but that is a flaw in the tool as it is such a common symptom.

R: While authors agree that ‘pain’ as a symptom would be useful in the GCSI tool, the FDA considers the ‘pain’ or ‘abdominal pain’ to be too general to be included in the tool. ‘Pain’ would have significant overlap with severe levels of ‘Bloating’, ‘Nausea’ and ‘Retching’ symptoms. Even the ‘Bloating’ symptom included in this study can only be used as an ‘exploratory item’, due to its overlap with ‘post-prandial fullness’ (Line 186-188). The communication between prominent gastroenterologists and the FDA regarding the inclusion/exclusion of symptoms in the ANMS-GCSI-DD manual is explored in reference 46.

Q: Line 43, reword 'with soluble fibres being easier to gastrically empty'

R: Changed to ‘with both types having an effect on gastric emptying’, a general statement in line with the citation (reference 12).

Q: Aims clear. Methods appropriate and very well described. Not sure the H2/CH4 BTs add much really though, and the inclusion of SIBO patients was curious, was this discovered post hoc?

R: Both participants were already diagnosed by their clinician with SIBO before their participation in the study (refer to line 314-317). The breath test in this study was not used for SIBO diagnosis, greater than >12 ppm H2 + CH4 merely a possible “positive” indicator for SIBO. A clarification has been added in text to remove ambiguity (Line 318-319).

Q: Additional tests to measure gastric emptying (13C  breath tests, plasma paracetamol) would have strengthened the study more than the OCTT part.

R: The authors agree that 13C breath tests / plasma paracetamol would have been better for the detailed characterisation of gastric emptying. Gastric emptying was not explored in detail due to the primary focus of this study (blood glucose and symptom effects) and resource limitations. 13C breath tests, plasma paracetamol and scintigraphy could be explored as research items for a future longer-term study involving ‘low-viscosity’ soluble fibres.

Q: What was the method of randomisation?

R: Randomisation of fibre intervention selection was by random number generator (Week 2), coin toss (Week 3) and last remaining (Week 4).

Q: Table 1, suggest report age and BMI to only 1 d.p.

R: Changed to 1 decimal point in Table 1 as indicated. Age also changed to one decimal point.

Q: Results are well described and interesting, though please remove word 'neatly' on line 213 though. Also, I don't think it's a paradox that the BG was lowest after water.

R: Phrasing has been changed removing ‘neatly’ from line 220 and ‘paradoxically’ from line 217.

Q: Discussion is balanced and fair. Not sure should yet be suggesting use of gastric alimetry as it is a new commercial device.

R: Reference to gastric alimetry removed as requested (Line 402 – 405). The technique would be worth revisiting at a later date if or when FDA approval is given, as it could have relevance in understanding the dynamics of gastric emptying.

Reviewer 2 Report

The topic of the article is certainly clinically relevant, however some major problems in the design of the study significantly reduce the value of presented data. The extremely low number of patients is the first critical point and the choice to use a "crossover" design does not solve this problem but introduces further variables that also make it extremely difficult to understand the description of the results.
Gastric scintigraphy is currently considered the ‘gold standard’ for gastroparesis diagnosis but the fact that enrolled participants had completed gastric scintigraphy within the preceding five years
introduces excessive variability in an already small population.
Taken as a whole, the design of the study and the description of the results are extremely confusing and make it very difficult to appreciate the value of some data which in themselves would be interesting.  

Author Response

Q: The topic of the article is certainly clinically relevant, however some major problems in the design of the study significantly reduce the value of presented data. The extremely low number of patients is the first critical point and the choice to use a "crossover" design does not solve this problem but introduces further variables that also make it extremely difficult to understand the description of the results. Taken as a whole, the design of the study and the description of the results are extremely confusing and make it very difficult to appreciate the value of some data which in themselves would be interesting.

R: Regarding the low sample size (N = 10), the authors would like to compare our study to recently published clinical studies investigating dietary fibres for IBS, constipation and diarrhoea where sample sizes typically range anywhere between 6 to 30. In these studies, either 1 or 2 dietary fibres are compared to a placebo (using a crossover design) and parameters such as resting blood glucose, insulin, CR-P and gut microbiome were tracked over a period of weeks. For conditions like IBS and others this approach is acceptable since participants do not feel any immediate adverse effects unlike in gastroparesis.

There is a lack of any major investigations regarding dietary fibre supplementation in gastroparesis patients and current investigations are often restricted to case studies in clinical settings (Reference 73). This study can be considered as a pilot study, a basis for a future long-term study as noted in lines 403-410. Therefore, a cautious approach, with a small sample size (N = 10), investigating the short-term effects of ‘low-viscosity’ soluble fibres was considered prudent. Also note that the suitability of several soluble dietary fibres for gastroparesis participants was carefully screened rheologically in vitro prior to clinical selection (Reference 29).

The authors feel that a randomized controlled crossover study in a small number of patients is ideally suited as baseline for future investigations. By studying three different fibres and the negative control in the same participant (crossover), and by normalizing the collected data (reducing variability), the changes and effects caused by the dietary fibre interventions becomes comparable both within the same participant (internal control) and other participants (since they also suffer from mild-to-moderate symptom gastroparesis).

There is statistical power of this type of analysis is enhanced, since major variables such as co-morbidities, medication and gender were carefully controlled. Participants with co-morbidities typically associated with gastroparesis other than Type-1 / Type-2 diabetes were excluded, reducing variability. The routine medications for each participant were the same across all four test weeks during the fasting period, minimizing variability. The participants were also instructed to avoid vitamin supplements as noted during fasting in Figure 1. It should also be noted that there was no gender-based variability since all participants were female (~80% of gastroparesis sufferers are females) with BMI and age is well correlated with symptom severity.

In summary, the results indicate that ‘low-viscosity’ soluble fibres PHGG and gum Arabic display glycaemic index lowering properties and blood glucose regulation similar to psyllium husk (positive control), while displaying symptom tolerability profiles similar to water (negative control). The ‘low-viscosity’ dietary fibres demonstrate properties that make them stand out as compromise candidates for dietary fibre supplementation in patients with mild-to-moderate symptom gastroparesis.

Q: Gastric scintigraphy is currently considered the ‘gold standard’ for gastroparesis diagnosis but the fact that enrolled participants had completed gastric scintigraphy within the preceding five years introduces excessive variability in an already small population.

R: Regarding gastric scintigraphy, clarifications have been added in section 2.2, noting that although the five-year diagnosis criteria was used for eligibility in this study, all participants in this study had completed a gastric scintigraphy test 12 months prior, either for the diagnosis of gastroparesis or to re-confirm their delayed gastric emptying and worsening symptoms. The cause for variability among the participants would be differences in their starting symptom severity, which is nullified by the internal control and data normalization. It should be noted that none of the participants were using enteral tube feeding (an exclusion criteria), which is typically used by sufferers with severe gastroparesis.

R: The following major changes have been made to the manuscript to provide clarity:

Title changed to “The short-term Effects and Tolerability of Low-Viscosity Soluble Fibre on Gastroparesis Patients: A Clinical Intervention Study”

Line 34: Added clarifying statement for proportion of female gastroparesis sufferers

Line 86-89: Statement regarding the primary and secondary endpoints have been added for clarity

Line 92: Changed to ‘randomised controlled crossover study’ for clarity

Line 134-144: Re-written for clarity regarding patient diagnosis and eligibility

Table 3: Added significant p-value tips for individual symptoms

Line 391-393: Provide clarity regarding the use of normalization of data

Line 402-405: Removed reference to gastric alimetry, as it is a relatively new investigative technique (Not FDA approved yet)

Line 406-408: Case study citation of dietary fibre supplementation in diabetic gastroparesis added

Round 2

Reviewer 2 Report

In the  new draft of the manuscript  no substantial changes have been made regarding the points previously underlined; for this reason I must confirm my negative opinion.

Author Response

The authors believe the questions raised by the reviewer regarding sample size and scintigraphy diagnosis time frame have been addressed in our Round 1 response. To address the reviewer finding the description of study design and results “extremely confusing”, we have further provided changes in the manuscript (highlighted) to concisely explain our methodology and findings.